# Navigating the New Reality: A Review of the Epidemiological, Clinical, and Microbiological Characteristics of *Candida auris*, with a Focus on Children

**DOI:** 10.3390/jof9020176

**Published:** 2023-01-28

**Authors:** Liat Ashkenazi-Hoffnung, Chen Rosenberg Danziger

**Affiliations:** 1Department of Day Hospitalization and Pediatric Infectious Diseases Unit, Schneider Children’s Medical Center, Petach Tikva 4920235, Israel; 2Sackler Faculty of Medicine, Tel Aviv University, Ramat Aviv, Tel Aviv 6997801, Israel; 3Department of Day Hospitalization, Schneider Children’s Medical Center, Petach Tikva 4920235, Israel

**Keywords:** antifungal resistance, *Candida auris*, children, echinocandin, invasive fungal infection, outbreak, neonatal intensive care unit

## Abstract

During the past decade, *Candida auris* emerged across the world, causing nosocomial outbreaks in both pediatric and adult populations, particularly in intensive care settings. We reviewed the epidemiological trends and the clinical and microbiological characteristics of *C. auris* infection, focusing on the pediatric population. The review is based on 22 studies, which included about 250 pediatric patients with *C. auris* infection, across multiple countries; neonates and premature babies were the predominant pediatric patient group affected. The most common type of infection reported was bloodstream infection, which was associated with exceptionally high mortality rates. Antifungal treatment varied widely between the patients; this signifies a serious knowledge gap that should be addressed in future research. Advances in molecular diagnostic methods for rapid and accurate identification and for detection of resistance may prove especially valuable in future outbreak situations, as well as the development of investigational antifungals. However, the new reality of a highly resistant and difficult-to-treat pathogen calls for preparedness of all aspects of patient care. This spans from laboratory readiness, to raising awareness among epidemiologists and clinicians for global collaborative efforts to improve patient care and limit the spread of *C. auris*.

## 1. Introduction

Included in the 2019 urgent threats report of the US Centers for Disease Control and Prevention (CDC) [1], *Candida auris* continues to spread throughout the world and cause nosocomial outbreaks in both pediatric and adult populations, particularly in intensive care settings [2,3,4,5,6]. The pathogen poses a serious challenge to healthcare systems due to its unique features, including extensive transmission among patients, persistence in hospital environments, misidentification by traditional laboratory methods, an antifungal-resistance profile and an association with high mortality rates [2,4,5,7,8,9]. The serious threat that *C. auris* poses has prompted public health agencies around the world to issue alerts to healthcare facilities on identifying and reporting incidences to health authorities [10,11,12,13,14]. 

The aim of this narrative review was to describe the epidemiological, clinical, and microbiological characteristics of *C. auris* infection, with a focus on pediatric patients. We have addressed specific groups affected and the spectrum of disease and outcome, and have provided practical recommendations for pediatricians for the identification, treatment and infection control of *C. auris*. 

For the purpose of this review, an electronic literature search was performed using PubMed, Google Scholar and clinicaltrials.gov, for reports on *C. auris* that were published through to 30 November 2022. Publications were reviewed and selected based on their quality and pertinence. Following exclusion of suspected repetitive studies, the search yielded 23 publications that reported *C. auris* incidents in children [15,16,17,18,19,20,21,22,23,24,25,26,27,28,29,30,31,32,33,34,35,36,37]. Twelve publications included both adult and pediatric patients [17,19,20,23,24,26,27,28,32,35,36,37]; of them, one did not provide specific details of the pediatric patients and was excluded [26]. Six provided limited specific details of the pediatric patients [17,19,24,27,37]. Five publications included data of other *Candida* species with limited data specific to *C. auris* infections [18,26,29,30,35]. Overall, 22 studies were included, involving about 250 *C. auris* cases in children. 

## 2. The Epidemiology of *C. auris*

### 2.1. Global Increase in Infections Caused by Non-Albicans Candida Species 

Recent decades have witnessed considerable changes in the distribution of *Candida* species that cause invasive candidiasis, including substantial increases in non-albicans *Candida* species, in both pediatric and adult populations [38,39,40]. This shift has been attributed to the widespread use of prophylactic antifungal drugs, such as azoles and echinocandins [41,42]. The shift from *C. albicans*, which is almost exclusively susceptible to all antifungals, to species that are more frequently resistant or tolerant to these drugs is concerning, and challenges clinicians due to the limited treatment options. Overall, *C. glabrata* has become significant in North America, Australia and most of Europe; while, *C. parapsilosis* is the dominant non-albicans species in South America, Japan and Spain [38,43,44]. However, species distribution has been shown to differ between pediatric and adult populations. In the pediatric population, *C. glabrata* and *C. krusei* are relatively rare in most geographical areas, whereas *C. parapsilosis* was reported as the predominant non-albicans species [40,45,46,47].

### 2.2. Emergence of Various Clones of C. auris 

Consistent with the above epidemiologic trends, independently and nearly simultaneously, a number of clones of *C. auris* have emerged in the past decade, in various geographical locations globally [3]. Genome-wide single nucleotide polymorphism (SNP)-based phylogenetic analyses have identified five major clades across the world: South Asian (I), East Asian (II), South African (III), South American (IV) and a novel clade from Iran (V) [3,15,48]. Excluding the notion of a single origin, clades differ considerably across regions, differing by 40,000 to 400,000 SNPs, and are almost identical within regions, differing by less than 70 SNPs. Moreover, clades have been shown to display unique clinical and microbiologic traits [49,50]. Until now, nosocomial outbreaks and invasive infections have been linked to clades I, III and IV of *C. auris*, while clades II and V have been primarily associated with ear colonization or infection, and not with invasive infections [15,48,50]. Clade I has also been associated with increased antifungal resistance compared to the other clades of *C. auris*; this includes a prominent feature of echinocandin resistance, mediated by the FKS1 mutation (S639Y) [51,52,53]. Phylogenetic analyses among pediatric patients identified clades I and V [22,48] [Dr. V. Anil Kumar, Amrita Institute of Medical Sciences, personal communication]. However, pediatric cases reported from South America and South Africa did not undergo phylogenetic analyses [16,17,29,31,32]. 

The reason for the recent nearly-simultaneous emergence of *C. auris* in multiple countries is unclear. Several explanations have been suggested. One hypothesis is that *C. auris* has long been present, but was not properly recognized microbiologically and was misidentified as a different species. This is supported by a few retrospective investigations of *Candida* species collections, such as a series of *C. haemulonii*, which identified *C. auris* isolates up to 1996 [19]. However, retrospective analyses of large-scale *Candida* isolate collections failed to identify *C. auris* in previous decades. For example, review of the SENTRY collection, of more than 20,000 *Candida* isolates from 39 countries from four broad geographic regions between 1997 and 2016, did not detect *C. auris* until 2009 [54]. Another hypothesis suggests that increased antifungal use in healthcare settings and in agriculture exerted selection pressure that favored the emergence of new drug-resistant *Candida* species. Examples of such are the general increase that has been observed in non-albicans *Candida* species and the specific emergence of echinocandin-resistant *C. glabrata* and azole-resistant *Aspergillus fumigatus* [55,56,57]. An additional plausible explanation relates to the interaction between global ecological changes and increasing mean global temperatures, and the distinctive biological properties of *C*. *auris* compared with other *Candida* species. These properties include thermotolerance, halotolerance and the ability to form resilient aggregates [58,59,60,61,62]. 

### 2.3. Timeline of C. auris Incidences

First reports of *C. auris* colonization and infection (non-invasive and invasive) increased in 2009 and 2011, respectively, among adults and children in Asia [20,63,64]. Nosocomial outbreaks, mostly in intensive care settings, were reported across Europe and Africa 2013–2015, and in North and South America, starting in 2016 [2,3,4,5,6,65]. The first incidences of transmission of *C. auris* among adults in Australia were reported in 2018 [66,67]. New introductions of *C. auris* are ongoing in a number of countries, as is the spreading from one country to another. Altogether, *C. auris* has been reported in six continents and at least 47 countries, with hundreds of new incidences detected each year worldwide [68,69]. According to the European CDC survey, 10 European countries have encountered patients colonized or infected with *C. auris* [70]. At present, 256 pediatric incidences have been reported from Venezuela, Colombia, Pakistan, Republic of Korea, India, Oman, Iran, Italy and the Gauteng province in South Africa [16,17,20,21,22,23,24,25,29,30,31,32,34,48] (Figure 1). The majority of pediatric incidences were reported from South America (114/256, 45%) and South Asia (67/256, 26%), in several nosocomial outbreaks. Thus, disease burden is currently lower in the pediatric population than the adult population. However, due to gaps in identification of *C. auris* and the lack of obligatory reporting in some countries, *C. auris* infections may be underreported in both children and adults. 

## 3. *C. auris* Microbiologic Identification 

Microbiologic identification of *C. auris* remains a serious challenge to healthcare systems, especially in developing countries. However, early and accurate microbiologic identification of the pathogen is essential for proper treatment and rapid implementation of infection control measures. Overall, laboratory capacities to identify *C. auris* have advanced considerably; however, not all countries are equally proficient. According to the European CDC survey conducted in 2018 and 2019, only 60% of laboratories were able to correctly identify a strain of *C. auris* [70]. Furthermore, European quality control trials confirmed the high rates of *C. auris* misidentification, reaching more than 40% [71,72]. 

### 3.1. Phenotypic Characteristics

On microscopy, *C. auris* is phenotypically indistinguishable from most other *non-albicans Candida* species [73]. It is a budding yeast that almost never produces pseudohyphae or hyphae. It grows well on Sabouraud’s dextrose agar as smooth white to cream-colored colonies; yet, in contrast to other *Candida* species, grows well at temperatures of 40–42 °C [2]. Similar to other non-albicans species, *C. auris* appears as pale purple or light pink on chromogenic agar, CHROMagar™ Candida (CHROMagar, Paris, France), and as blue colonies on CHROMagar™ Candida Plus (CHROMagar, Paris, France) [74]. The use of growth characteristics on chromogenic agar, supplemented with Pal’s medium, has been suggested as a low-cost method to differentiate between isolates of *C. auris* and *C. haemulonii* [75]. However, in general, chromogenic agars should not be considered as final identification of *C. auris*. Growth of non–*albicans* species on chromogenic agar should prompt sub-culturing onto Sabouraud’s agar and subsequent identification by other reliable methods. 

### 3.2. Diagnostic Biochemical Assays

Further on the unreliability of conventional phenotypic methods, the reliance on traditional methods that are based on biochemical assays may also lead to misidentification, due to a lack of reference databases [65,71,73,76,77]. Examples of such biochemical assays are the VITEK^®^ 2 (bioMérieux), BD Phoenix (Becton Dickinson), API^®^ 20C AUX (bioMerieux), API^®^ Candida and MicroScan (Beckman Coulter). *C. auris* is most frequently misidentified as *C. haemulonii*, but also as *C. famata*, *C. lusitaniae*, *C. sake*, *C. catenulata*, *C. guilliermondii*, *C. parapsilosis*, *C. tropicalis*, *C. albicans* and *Rhodotorula glutinis* (Table 1) [77,78,79,80]. Indeed, several case series reported that the use of VITEK^®^ 2 and API 20C initially misidentified *C. auris* in children [16,17,31,32] (Table 2). Therefore, reports of the abovementioned uncommon species by these systems should raise suspicion for *C. auris*. Moreover, high vigilance is necessary when the incidence of an unidentified *Candida* species increases, with or without resistance to fluconazole or amphotericin B, and in instances of isolated yeasts from patients with a high risk of *C. auris*. The latter includes residents of countries with extensive transmissions of *C. auris* [i.e., South Africa, South America (Colombia, Venezuela and Panama) and South Asia (India and Pakistan) [81]]. 

### 3.3. Recommended Diagnostic Methods for C. auris

Correct *C. auris* identification requires specialized laboratory methodology, such as the use of matrix-assisted laser desorption ionization-time of flight mass spectrometry (MALDI-TOF MS), including reference *C. auris* spectra in the database [84,85,86]. Currently available accurate databases include the FDA-approved MALDI Biotyper CA System library (Version Claim 4) and their “research-use only” libraries [Versions 2014 (5627) and more recent] for the Bruker Biotyper MALDI-TOF, and the FDA-approved IVD library (v3.2) or “research-use only” library (Saramis Version 4.14 database with Saccharomycetaceae update) for bioMérieux VITEK (MALDI-TOF) MS RUO [87]. Low awareness was reported regarding the need to update the libraries of Dutch clinical microbiological laboratories on *C. auris* spectra by means of MALDI-TOF MS [71]. Clearly, as databases are updated, accurate identification will become more feasible. Of note, *C. auris* is currently not among the five species included in the T2Candida Panel [88]. 

Although less available for routine identification, molecular sequencing using polymerase chain reaction (PCR) assays offers definitive *C. auris* identification. Several molecular-based assays have been developed, including conventional PCR, real-time PCR, T2 magnetic resonance and loop-mediated isothermal amplification (LAMP) assays [89,90,91]. In contrast to biochemical automated systems and MALDI-TOF MS, which are culture dependent, DNA can be isolated directly from patients’ specimens without the need for a culture. Therefore, molecular-based assays can provide rapid results, and carry the potential for high-throughput screening of surveillance samples in an outbreak setting. Moreover, molecular sequencing of ribosomal DNA loci, such as the internal transcribed spacer (ITS1, ITS2) region or the D1/D2 region of large subunits (LSU), enables differentiating between geographic clades [92]. 

## 4. Clinical Spectrum of *C. auris* Infection

### 4.1. General Clinical Characteristics of C. auris Infection, with a Focus on Children

The spectrum of *C. auris* infection ranges widely from superficial skin infection to invasive disease. *C. auris* was described as progressing from colonization to invasive infection in 4–25% of affected adults [93,94]. Common sites of *C. auris* colonization described in adults were the skin, especially the groin and axilla areas, and mucosal surfaces, i.e., the genitourinary tract and the gastrointestinal and respiratory tracts (oropharynx, nose, ears) [4,5,65,95,96]. In children, however, asymptomatic colonization was rarely described. Colonization was reported in a neonate born to a colonized mother; the skin (axilla), eyes and ears were involved [21]. Also, progression from colonization to infection was not clearly described. The lack of pediatric reports of colonization may be due to the decreased screening rates, consequent to the relatively-lower rate of nosocomial outbreaks. Nevertheless, in a point prevalence survey for *C. auris* colonization in a pediatric long-term transitional care hospital in the United States, *C. auris* was not identified [97]. This is despite a high prevalence of *C. auris* among adult patients in health care settings of similar acuity in the region.

According to the 22 publications of *C. auris* infection, comprising 256 children [15,16,17,18,19,20,21,22,23,24,25,27,28,29,30,31,32,33,34,35,36,37], reviewed herein, the most common type of invasive infection was bloodstream infection (94%, 194/206 patients with available data on the infection site) (Table 3 and Appendix A). The duration of candidemia was not reported in most studies and was available for only seven patients; the duration ranged between 7 and 11 days [16,20,36]. Other sites of infection included meningitis, endocarditis, intravascular infection, peritonitis, urinary tract infection, skin abscess and otitis [16,25,34]. The associated mortality rates reported ranged from 0%, to as high as 80%. Most series reported morality rates of ~40%. However, not all the mortality reported was attributable to *C. auris* infection. A recurrent episode was described in a five-month old infant readmitted with thrombosis of a systemic-to-pulmonary artery shunt, several months after the initial candidemia [16]. Antifungal treatment varied between studies. A number of studies treated invasive infections with antifungal combination [16,22,25,34]. Due to the small number of patients, conclusions could not be drawn regarding differences in mortality rates according to antifungal regimens. Patient age ranged from 1 day to 14 years; male predominance was described in most series (8/13 with available data on sex). The pediatric patient groups affected by *C. auris* infection primarily comprised neonates and children born prematurely. Accordingly, 12 of 16 case series in children involved neonatal intensive care units (ICUs), and 70 of 214 patients (33%) were neonates or children born prematurely. Other well-established risk factors for candidemia in children were also present. Of the 135 patients with available data on underlying conditions, 94 (70%) had a central venous catheter, 82 (61%) were on total parenteral nutrition, 54 (40%) had been exposed to broad-spectrum antibiotics, 29 (22%) had undergone a prior surgical procedure and 31 (23%) had congenital or acquired immune deficiency. One pediatric patient was identified during the COVID-19 pandemic [24]. 

### 4.2. Clinical Characteristics of C. auris Infection Compared to Other Candida Species 

The colonization sites, clinical spectrum of disease, characteristics of affected patients and the risk factors for invasive disease were similar between infections caused by *C. auris* and by other *Candida* species [40,47,98]. However, prior colonization as a risk factor for developing candidemia was not clearly described for *C. auris*, as described for other *Candida* species [40]. 

Overall mortality appeared higher in patients with *C. auris* infections (~40%) than with candidemia caused by *C. albicans* or non-*albicans* species (12–20%), as reported in historical pediatric cohorts [40,47,99,100]. Nonetheless, a nationwide Indian study of candidemia in children in intensive care settings described higher mortality with *C. auris* only among non-neonates, whereas among neonates, mortality was similar for *C. auris* (33%), *C. parapsilosis* (40%) and *C. albicans* (40%) [33]. 

Another unique characteristic of *C. auris* compared to other *Candida* species is the involvement in nosocomial outbreaks. This was rarely described with other *Candida* species, with the exception of *C. parapsilosis* [101,102,103]. This feature may be related to the mutual propensity of *C. auris* and *C. parapsilosis* to colonize the skin and enable person-to-person spread.

## 5. *C. auris* Antifungal Resistance and Therapeutic Options 

### 5.1. C. auris Susceptibility Profile

One of the main reasons for global concern about the spread of *C. auris* is its susceptibility profile, which limits treatment options. Most isolates of *C. auris* are resistant to fluconazole, are often cross-resistant to other azoles, and have variably elevated minimum inhibitory concentration (MICs) for amphotericin B [51,76,78,93,104,105,106,107]. Echinocandins have the lowest MICs for *C. auris* of all systemic antifungal classes, but resistance to these drugs has been described [3,6,51,52,83,106,107,108,109]. Longitudinal data suggest that echinocandin resistance rates are increasing [110].

Despite the above, *C. auris*-specific susceptibility breakpoints have not been established. The susceptibility categorization of *C. auris* isolates is based on tentative MIC breakpoints that were suggested by the CDC, based on those established for closely related *Candida* species and on expert opinion [82]. The available CDC tentative MIC breakpoints are as follows: fluconazole ≥32 µg/mL, amphotericin B ≥2 µg/mL (E-test values of 1.5 rounded up to 2), caspofungin ≥2 µg/mL and micafungin ≥4 µg/mL [82]. Epidemiologic cutoff values have also been suggested [83]. 

Based on these tentative MICs, susceptibility data from the United States and the United Kingdom showed resistance of 90–100% of *C. auris* isolates to fluconazole, 20–30% to amphotericin B and 5–10% to echinocandins [13,111]. Higher resistance rates to amphotericin B, by more than 60%, were recorded in an analysis of 277 clinical isolates in an outbreak of *C. auris* in New York between2016 and2018 [112]. A few isolates in a number of countries have demonstrated elevated MICs to multiple classes of antifungal agents [3,35,78,109,110,113]. Thus, while pan-resistant *C. auris* still appears rare, its emergence is concerning. Susceptibility to antifungals varied widely among studies in children. MICs for antifungal drugs were reported in 13 studies in children (Table 2). As described in adults, most *C. auris* isolates in pediatric series were susceptible to echinocandins, with low MIC values; however, for fluconazole and amphotericin B, MICs were variable. The unexpected, relatively low rates of resistance to fluconazole were mainly reported from Colombia and the Republic of Korea, and may be related to differences in phylogenic characteristics or local azole use. 

A number of studies have described the molecular mechanisms in *C. auris* that result in antifungal resistance and clinical failures of azoles and echinocandins. Resistance to azoles was shown to be mediated by mutations in ERG11 (F126L, Y132F and K143R) [3,51,106,114,115] and in CDR1 (V704L) [115]; and resistance to echinocandins, by mutations in FKS1 (S639P, S639F, S639Y, F635C, S635P and S635T) [51,52,115,116,117,118]. Analysis of pan-resistant *C. auris* strains suggested a fitness cost in some strains [118]. Recently, real-time PCR was developed for the identification of mutations in *C. auris* ERG11 and FKS1 genes. As with other molecular testing, this test has the advantage of rapid detection of *C. auris* antifungal resistance directly from clinical swabs [119]. Beyond these limited mutations, the genetic basis for *C. auris* resistance remains unclear. One study aimed at delineating the impact of ERG11 mutations (F125L, Y132F and K143R) on fluconazole susceptibility in *C. auris* clinical isolates with a Cas9-mediated transformation system [120]. The conclusion was that even though all these mutations contribute to fluconazole resistance, none alone are sufficient to confer clinical resistance and cannot explain the significantly elevated MICs among clinical isolates of *C. auris*.

### 5.2. Recommendations for Treatment of C. auris 

Based on the abovementioned MIC data, concern for resistance to azoles and amphotericin B led the CDC and Public Health England to recommend echinocandins as first-line treatment of *C. auris* infections [13,111]. However, thus far, a correlation between in vitro susceptibility testing and clinical outcomes has not been discerned. Observational data from nosocomial outbreaks show high rates of mortality among patients infected with *C. auris*, regardless of the choice of antifungal agent, among both adults [3,4,36,62] and children [16,25,31,34]. Furthermore, previous exposure to both fluconazole and echinocandins was consistently associated with an increased risk of *C. auris* infection [3,4,121]. This suggests that these drugs exert selective pressure that favors the survival of *C. auris*. Breakthrough *C. auris* infection or late complications upon echinocandin therapy [4,5,105], and the emergence of echinocandin-resistant *C. auris* strains during treatment [93,112,116,122] were also noted. Preliminary animal studies showed that echinocandins did not affect survival rates of neutropenic mice with hematogenic *C. auris* infection, whereas amphotericin B increased survival by 50% [123]. These observations further underscore the need for additional clinical data to guide antifungal treatment. 

Until efficient clinical data are available and optimal antifungal treatment is defined, current recommendations suggest initial empiric treatment of *C. auris* infections with an echinocandin, for infants aged two months and older [13,111]. However, because of the potential for rapid development of resistance during therapy, follow-up cultures and repeated susceptibility testing should be conducted, especially in patients treated with echinocandins. Treatment should be switched to liposomal amphotericin B if clinical response is inadequate or candidemia >5 days persists during treatment with echinocandins [111]. Initial treatment with liposomal amphotericin B should be considered in patients with prior prolonged exposure to echinocandins, for whom echinocandin resistance is a concern.

For neonates and infants under the age of two months, the initial treatment of choice is amphotericin B deoxycholate. If a patient does not respond to this drug, liposomal amphotericin B can be considered. In exceptional circumstances, when the central nervous system involvement has been ruled out, treatment with echinocandins may be cautiously considered [124,125] (Table 4).

In parallel to empiric therapy, all *C. auris* isolates should undergo antifungal susceptibility testing according to guidelines of the Clinical Laboratory Standard Institute and the European Committee for Antimicrobial Susceptibility Testing (EUCAST). MICs were very similar in these two guidelines [83]. As they are based on the evaluation of growth inhibition, current conventional antifungal susceptibility testing methods, including reference and commercial types, are limited by a high turnaround time of 24 to 72 h, from positive culture of the clinical sample to susceptibility results. A number of innovative methods with a short time to results are currently under development and evaluation, with potential for guiding earlier definitive antifungal treatment. These include methods based on MALDI-TOF MS, flow cytometry and computed imaging [128]. A recent systematic review and meta-analysis showed high-level diagnostic accuracy of antifungal susceptibility testing based on MALDI-TOF MS [129]. Of the twelve studies reviewed, one study specifically evaluated echinocandin susceptibility testing in *C. auris* derived from Sabouraud’s dextrose agar and blood culture bottles [130]. Using the MALDI Biotyper antibiotic susceptibility test-rapid assay (MBT ASTRA), the study demonstrated the applicability of this method for rapid susceptibility testing in *C. auris*. Molecular testing of mutations associated with antifungal resistance, such as the mutations in *C. auris* ERG11 and FKS1 genes mentioned earlier, is another evolving alternative to conventional susceptibility testing [119]. It bears the advantage of rapid detection of resistance directly from clinical swabs and the ability to concomitantly detect resistance to multiple classes of antifungals. Nevertheless, it is restricted to known mutations. Next-generation sequencing and whole-genome approaches may overcome this limitation in the near future.

Treatment of pan-resistant *C. auris* strains is a clinical challenge. Combination antifungal treatment yielded mixed results in laboratory testing and has not been systematically evaluated in clinical settings [115,118,123,131,132,133,134,135]. Some in vitro studies showed that effective treatments against pan-drug-resistant *C. auris* are flucytosine combinations with amphotericin B, azoles or echinocandins, or amphotericin B and echinocandin combined [118,134,135]. However, other studies did not find such results [115,136]. Investigational drugs against *C. auris* may be considered for patients with echinocandin- or pan-resistant isolates. Albeit not yet investigated among pediatric patients, a number of new antifungal drugs are currently in various stages of development and clinical trials. Ibrexafungerp (SCYNEXIS, formerly known as SCY-078, Jersey City, NJ, USA) is a first-in-class oral glucan-synthase inhibitor, that was shown to be active against azole-resistant and echinocandin-resistant *C. auris* [137,138,139,140]. This drug is currently being assessed in an open-label, single-arm, phase 3 trial in adults aged 18 years and older, for the treatment of documented *C. auris* infections (NCT03363841). In 2021, Ibrexafungerp received regulatory approval for its first product from the US Food and Drug Administration, based on trials evaluating treatment of vulvovaginal candidiasis. APX001 (Amplyx Pharmaceuticals, San Diego, CA, USA) is another first-in-class drug that targets a novel pathway—glycosylphosphatidylinositol glycolipid biosynthesis. In vitro and animal studies showed APX001 to be active against *C. auris* [141,142,143,144]. This drug was assessed in an open-label, single-arm, phase 2 trial in adults aged 18 years and older, for the treatment of invasive candidiasis caused by *C. auris* (NCT04148287).

In instances of *C. auris* isolation from non-invasive sites, such as the skin, rectum or respiratory tract, antifungal treatment is not recommended [111]. Similar to recommendations for other *Candida* species, treatment is generally only indicated if clinical disease is present. However, infection control measures should be used for all patients with *C. auris*, regardless of the source of the specimen.

### 5.3. Recommendations for Prophylaxis

In settings of high rates of *C. auris*, some authors have advised for antifungal prophylaxis for low-birth weight preterm neonates with echinocandins, as an alternative to the standard prophylaxis with fluconazole [145]. This recommendation is due to the prevalent fluconazole resistance of *C. auris* [21,34]. A small comparative clinical study reported that micafungin compared to fluconazole prophylaxis against fungal infections in extremely low-birthweight infants was associated with a decreased incidence of *C. albicans* infections [146]. Additionally, safety and pharmacokinetics of micafungin were previously assessed in very low birthweight infants [147,148,149]. Thus, in the setting of a neonatal ICU outbreak, micafungin prophylaxis can be considered for high-risk populations (Table 4). However, as mentioned earlier, concern arises that echinocandin use may exert selective pressure favoring the emergence of *C. auris*. 

## 6. Infection Control Measures against *C. auris*

The remarkable widespread horizontal transmission of *C. auris* between patients in healthcare facilities is a source of nosocomial outbreaks [5,96]. This is likely due to the capability of *C. auris* to colonize the skin of patients and healthcare personnel [95,150], and to survive outside the host on environmental surfaces and medical equipment for long periods of time [151]. Moreover, the pathogen is resistant to commonly used disinfectants, such as quaternary ammonium compounds [7]. Outbreaks of *C. auris* have also been reported in designated adult COVID-19 units in India, Colombia, Mexico and the US [26,152,153,154,155]. One *C. auris* infection was identified in a COVID-positive pediatric patient; however, nosocomial transmissions among children in such a setting was not described [24]. Vertical transmission was suspected, from a *C. auris* colonized mother through vaginal delivery to her offspring; however, environmental and maternal transmission could not be discriminated [21]. 

Due to the potential of *C. auris* for calamitous nosocomial outbreaks, recommendations for infection control measures have been issued [13,14,156]. The extent of practice depends on the local prevalence of *C. auris* and the burden of disease [22]. For instance, in a pilot study of screening for *C. auris* in ICUs in England, that had no previous incidences of *C. auris*, colonization was not detected. This led the authors to recommend against widespread screening for *C. auris* in ICUs in England, at present, and in favor of limiting screening to high-risk individuals based on local risk assessment [157]. This is in line with the recommendations of health authorities and The Infection Prevention and Control working group of the International Society of Antimicrobial Chemotherapy for healthcare workers on infection prevention and control measures for *C. auris* at inpatient healthcare facilities [158]. Screening is advised in units with a new identification of *C. auris* or with ongoing patients with *C. auris* [14]. Steps advised for controlling *C. auris* are presented in Figure 2. Preliminary steps include raising awareness and providing education to all healthcare personnel. 

Infection control precautions are advised when screening identifies a patient with *C. auris* or for patients with a clinical *C. auris* infection. Standard infection control measures should be rapidly applied and continued until patient discharge. This is based on persistent colonization of patients with *C. auris* in surveillance studies [159,160]. Investigations have also shown high positivity rates for *C. auris* from environmental samples, such as collected from bedrails, windowsills and shared medical equipment [160]. Therefore, after discharge, reusable medical equipment and rooms should be cleaned and disinfected using chlorine-based disinfectants at a concentration of 1000 ppm, hydrogen peroxide or other disinfectants with documented fungicidal activity. Quaternary ammonium compound disinfectants should be avoided [161]. Detection of a *C. auris* infection should prompt an epidemiological investigation and screening of close-contact patients for *C. auris* carriage. Suggested screening sites by the CDC are the groin and axilla, bilaterally, as these sites have been identified as the most common and consistent sites of colonization in adults. Other sites considered for sampling are: urine, and the nose, throat and rectum [13]. In community settings, local authorities are advised to exclude the attendance of children with *C. auris* wound infections from daycare until drainage from wounds, or skin and soft tissue infections are contained [162]. 

In addition to the above, health authorities should recommend active prospective surveillance for *C. auris*, namely routine notification of *C. auris* by laboratories and healthcare professionals [163]. The inclusion of *C. auris* in the national list of statutory notifiable causative organisms is strongly advised. Some experts have suggested laboratories to review past records of suspected *Candida* species [164]. Consistent gathering of epidemiological data at national and international levels will enable informed and coordinated risk management actions by public health authorities.

## 7. Conclusions

Although *C. auris* infections are still relatively rare among neonates and children, its worldwide emergence in multiple countries and various continents represents a new reality that calls for preparation of all aspects of patient care. This spans laboratory readiness for adequate detection to high vigilance of clinicians for any unidentified or rarely encountered *Candida* species, in order to rapidly implement infection control measures. Associations of *C. auris* with nosocomial outbreaks in neonatal ICUs, invasive infections, high-level antifungal resistance and high mortality rates, highlight the importance of global collaborative efforts to raise awareness and limit its spread. Future research should address knowledge gaps in appropriate antifungal treatment.

## Figures and Tables

**Figure 1 jof-09-00176-f001:**
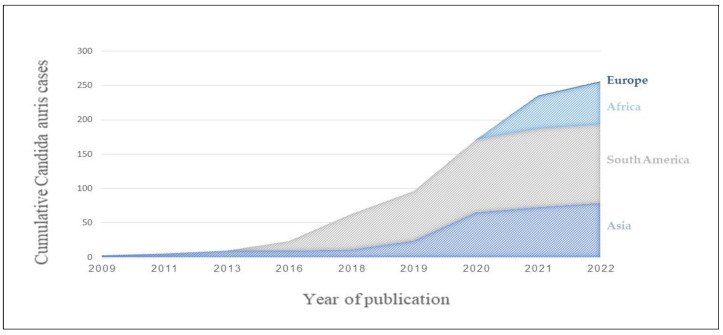
Confirmed cumulative *C. auris* incidences in children, by official year of publication.

**Figure 2 jof-09-00176-f002:**
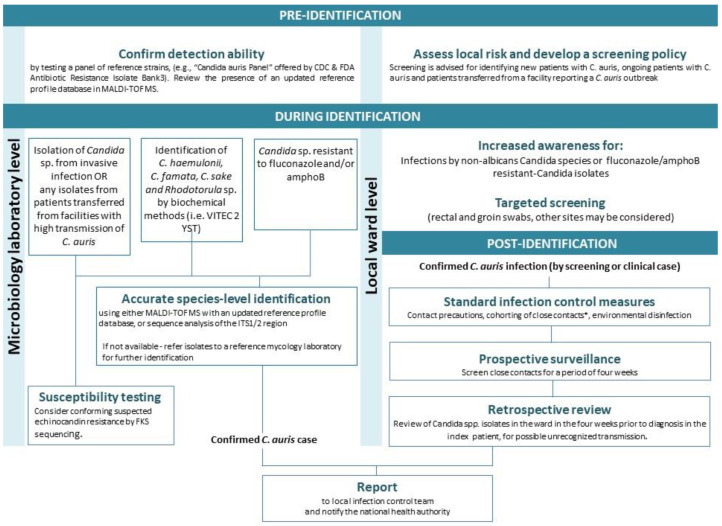
Recommended steps for preparedness and control of *C. auris.* The steps are based on recommendations of the US Centers of Disease Control and Public Health England [13,156]. See the text for complimentary recommendations. amphoB, amphotericin B; MALDI-TOF MS, Matrix-assisted laser desorption/ionization time-of-flight mass spectrometry; ITS1/2, internal transcribed spacer. * Close contacts can be de-isolated after three consecutive negative screens at least 24 h apart.

**Table 1 jof-09-00176-t001:** Biochemical platforms that may misidentify *C. auris*.

Method of Identification	Potential Misidentification [77]
VITEK^®^ 2 YST (bioMerieux)	*C. haemulonii*, *C. lusitaniae* [80], *C. famata* [78], *C. pelliculosa* [16]
BD Phoenix™ (Becton Dickinson)	*C. haemulonii*, *C. catenulata*
API^®^ 20C AUX (bioMerieux)	*C. sake*, *Rhodotorula glutinis*
API^®^ Candida	*C. famata* [79]
MicroScan (Beckman Coulter)	*C. famata*, *C. guilliermondii*, *C. lusitaniae*, *C. parapsilosis*, *C. tropicalis* [79], *C. albicans* [79]

**Table 2 jof-09-00176-t002:** Microbiologic characteristics of *C. auris* isolates in children. (**a**) Microbiologic identification of *C. auris* isolates. (**b**) Susceptibility profile of *C. auris* isolates.

(a)								
Country	Number of Cases	Method of Reliable Identification		Initial Misidentification		Method of Misidentification		Reference
Colombia	34	MALDI-TOF MS		*C. haemulonii*, *C. guilliermondii*, *C. albicans*, *C. parapsilosis*, *Rhodotorula rubra*		BD Phoenix, microscan		Berrio et al. [31]
Colombia	39	MALDI-TOF MS		*C. haemulonii*, *C. albicans*, *C. guilliermondii*, *C. parapsilosis*, *R. rubra*		NA		Escandon et al. [17]
Colombia	8	50% MALDI-TOF MS 50% presumed *C. auris* due to susceptibility profile		*C. haemulonii*, *C. pelliculosae*		VITEK2		Alvarado-Socarras et al. [16]
Venezuela	13	ITS sequencing		*C. haemulonii*		VITEK2		Calvo et al. [32]
Iran	1	MALDI-TOF MS, rDNA sequencing, WGS		non-*albicans Candida*		Phenotypic characterization		Abastabar et al. [15]
Iran	1	MALDI-TOF MS, ITS sequencing		none		none		Mirhendi et al. [22]
India	17	Sequencing (2016), VITEK with ID system software version 8.01 software update (2017)		*C. haemulonii*, *C. duobushaemulonii*		VITEK		Chandramati et al. [34] *
India	5	VITEK2 MALDI-TOF (VITEK MS) and PCR		NA		NA		Ramya et al. [25]
India	5	ITS and D1/D2 region sequencing		*C. haemulonii*, *C. famata*, *C. sake*		VITEK2, API20C		Chowdhary et al. [36]
India	1	MALDI-TOF and ITS or D1/D2 region sequencing		NA		NA		Kaur et al. [18]
India	22	ITS sequencing		NA		NA		Chakrabarti et al. [33]
Bangladesh	3	ITS sequencing		NA		NA		Sathi et al. [28]
Pakistan	1	Profile numbers 2,000,130, 2,000,173, 2,102,173, 6,102,173 on API 20C AUX in conjunction with phenotypic characteristics and susceptibility profile		NA		NA		Moin et al. [44]
North Korea	3	ITS and D1/D2 region sequencing		*C. haemulonii*, *R. glutinis*		VITEK2 YST and API 20C, respectively		Kim et al. [45]
Republic of Korea	2	ITS sequencing		*C. haemulonii*, *R. glutinis*		VITEK2 and API20C		Lee et al. [34]
Oman	2	MALDI-TOF MS ITS sequencing		*C. haemulonii*, *C. famata*, *R. glutinis*		BD Phoenix and API AUX 20C		Mohsin et al. [23]
Italy	1	MALDI-TOF MS		NA		NA		Mesini et al. [43]
Gauteng province	47	NA		NA		NA		Shuping et al. [29]
**(b)**								
**Country**	**Number of Cases**	**Susceptibility Profile** **% of Resistance (available MIC data, μg/mL)**						**Reference**
		**FLC**	**VRC**	**CAS**	**MFG**	**AFG**	**AMB**	
Colombia	34	15%	NA	0%	0%	8%	54%	Berrio et al. [31]
Colombia	39	30%	NA	NA	NA	1%	22%	Escandon et al. [17]
Colombia	8	16.7% (MIC range <2–≥64)	0% (MIC range ≤ 0.12–1)	0% (MIC range ≤ 0.25)	0% (MIC range < 0.12)	NA	100% (MIC range 8–≥64)	Alvarado-Socarras et al. [16]
Venezuela	13	100% (MIC range > 64)	100% (MIC 4)	NA	NA	0% (MIC range 0.06–0.125)	NA (MIC range 1–2)	Calvo et al. [32]
Iran	1	0% (MIC 16)	0% (MIC 0.125)	NA	0% (MIC 0.031)	0% (MIC 0.016)	0% (MIC 0.5)	Abastabar et al. [15]
Iran	1	100% (MIC > 64)	0% (MIC 0.25)	0% (MIC 0.5)	NA	0% (MIC 1)	0% (MIC 1)	Mirhendi et al. [22]
India	17	100%	0%	NA	0%	NA	NA	Chandramati et al. [34] *
India	5	NA	0%	NA	0%	0%	NA	Ramya et al. [25]
India	5	100% (MIC range 16–64)	0% (MIC range 0.125–1)	0% (MIC range 0.125–0.25)	0% (MIC range 0.06–0.125)	0% (MIC range 0.125–0.5)	0% (MIC range 0.25–1)	Chowdhary et al. [36]
India	1	100% (MIC 64)	0% (MIC 0.5)	0% (MIC 0.5)	NA	0% (MIC-2)	100% (MIC-4)	Kaur et al. [18]
India	22	55% (MIC_50_-8–64)	5% (MIC_50_-0.38–1)	5% (MIC_50_- 0.5–0.75)	0% (MIC_50_- 0.09–1)	0% (MIC_50_- 0.12–0.25)	5% ** (MIC_50_-0.12–0.5)	Chakrabarti et al. [33]
Bangladesh	3	100% (MIC 64)	33%	NA	NA	NA	100% (MIC 4)	Sathi et al. [28]
Pakistan	1	100%	NA	0%	NA	0%	NA	Moin et al. [44]
North Korea	3	47% (MIC range 2–128)	NA	0% (MIC range 0.125–0.25)	0% (MIC 0.03)	NA	33% (MIC range 0.38–1.5)	Kim et al. [45]
Republic of Korea	2	33% (MIC range 2–128)	33% (MIC range 0.03–1)	0% (MIC 0.06)	0% (MIC 0.03)	NA	0% (MIC range 0.5–1)	Lee et al. [34]
Oman	2	100% (MIC 64)	0% (MIC 0.5)	NA	0% (MIC range 0.125–0.25)	0% (MIC range 0.125–0.5)	50% (MIC range 1–2)	Mohsin et al. [23]
Italy	1	100% (MIC > 256)	NA	0% (MIC 0.12)	0% (MIC 0.12)	0% (MIC 0.25)	0% (MIC 1)	Mesini et al. [43]
Gauteng province	47	90% (MIC range 16–256)	NA	NA	0% (MIC range 0.03–1)	0% (MIC range 0.06–0.5)	0% (MIC range 0.003–1)	Shuping et al. [29]

MIC, Minimal inhibitory concentration; FLC, Fluconazole; VRC, Voriconazole; CAS, Caspofungin; MFG, Micafungin; AFG, Anidulafungin; AMB, Amphotericin B; NA, not available; MALDI-TOF MS, Matrix-assisted laser desorption ionization-time of flight mass spectrometry; ITS, internal transcribed spacer; rDNA, ribosomal DNA; WGS, whole genome sequencing; PCR, polymerase chain reaction; MIC_50/90_, 50%/90% minimum inhibitory concentration. MICs were interpreted using the CDC tentative breakpoints [82], or for voriconazole, using suggested epidemiological cutoffs [83]. * Dr. V. Anil Kumar, Amrita Institute of Medical Sciences, personal communication]. ** Defined as MIC > 1.

**Table 3 jof-09-00176-t003:** Demographic and clinical characteristics of children with *C. auris* infection or colonization.

Continent	Country	Number of Cases	Age, Mean	Male Sex	BSI	Underlying Conditions					Treatment			Mortality	Reference
						Preterm	CVC	TPN	Surgery	Immunodeficiency ^1^	Azole	Echinocandin	AmphoB		
South America	Colombia	34	NA	64%	100%	26%	82%	56%	15%	44%	29%	21%	47%	41%	Berrio et al. [31]
	Colombia	39	NA (19%) aged < 1 yr)	NA	NA	NA	NA	NA	NA	NA	NA	NA	NA	NA	Escandon et al. [33]
	Colombia	8 ^2^	16 d	NA	50%	13%	38%	NA	75%	NA	38%	88%	0%	38%	Alvarado-Socarras et al. [30]
	Colombia	12 ^3^	34 d (median)	75%	100%	50%	100%	92%	75%	33%	NA	NA	NA	42%	Armstrong et al. [27]
	Venezuela	13	<2 m, one aged 14 yr	46%	100%	61%	100%	NA	46%	NA	85%	69%	23%	31%	Calvo et al. [32]
Asia	Iran	1	14 yr	0%	0%	0%	0%	0%	0%	0%	0%	0%	0%	0%	Abastabar et al. [22]
	Iran	1	2.5 yr	100%	0%	0%	NA	NA	NA	100%	100%	0%	100%	NA	Mirhendi et al. [29]
	India	17	19 d	70%	88%	88%	100%	94%	47%	NA	71%	41%	53%	41%	Chandramati et al. [42]
	India	1	NA	NA	100%	NA	NA	NA	NA	NA	NA	NA	NA	NA	Kaur et al. [18]
	India	5	9 d	60%	100%	100%	100%	1000%	NA	NA	60%	100%	0%	80%	Ramya et al. [41]
	India	5	2 yr	20%	100%	40%	80%	NA	20%	80%	0%	20%	60%	40%	Chowdhary et al. [79]
	India	3	<1 m	NA	NA	NA	NA	NA	NA	NA	NA	NA	NA	67%	Singh et al. [30]
	India	22	NA (27% < 1 m)	NA	100%	18%	44%	NA	25% ^4^	NA	NA	NA	NA	41%	Chakrabarti et al. [33]
	Bangladesh	13	<1 m	NA	100%	NA	NA	NA	NA	NA	NA	NA	NA	NA	Dutta et al. [37]
	Bangladesh	3	10 d	100%	100%	NA	NA	NA	NA	NA	100%	0%	0%	67%	Sathi et al. [28]
	Pakistan	1	NA	100%	100%	NA	100%	NA	NA	NA	NA	NA	NA	0%	Moin et al. [44]
	North Korea	3	NA	NA	0%	NA	NA	NA	NA	NA	NA	NA	NA	NA	Kim et al. [45]
	Republic of Korea	2	1 yr	50%	100%	0%	50%	100%	50%	50%	100%	0%	100%	50%	Lee et al. [34]
	Oman	2	1 yr	100%	100%	NA	NA	NA	NA	50%	NA	NA	NA	NA	Mohsin et al. [23]
Europe	Italy	1	1 d	0%	0%	100%	NA	NA	NA	NA	0%	0%	0%	100% ^5^	Mesini et al. [43]
South Africa	Gauteng province	47	NA (15% < 1 m)	NA	100%	NA	NA	NA	NA	NA	NA	NA	NA	NA	Shuping et al. [29]
	Gauteng province	15	NA (93% < 1 m)	NA	100%	NA	NA	NA	NA	NA	NA	NA	NA	NA	Chibabhai et al. [35]

BSI, blood stream infection; CVC, central venous catheter; TPN, total parenteral nutrition; AmphoB, amphotericin B; NA, not available. ^1^ Including: congenital immunodeficiency [Chronic Granulomatous Disease (CGD)], neutropenia, malignancy, chemotherapy, hemophagocytic lymphohistiocytosis (HLH), corticosteroid use. ^2^ Four included in [17] [Dr. Rodriguez-Morales, Universidad Tecnológica de Pereira, Colombia, personal communication]. ^3^ Additional eight aged 1–18 years without separate clinical data. ^4^ In neonates and non-neonates, respectively. ^5^ Unrelated to *C. auris* isolation.

**Table 4 jof-09-00176-t004:** Recommended antifungal treatment and prophylaxis for children with *C. auris* infection.

Age Group	Preferred Treatment Regimen	Dosing	Alternate Regimen ^1^	Dosing
Neonates and infants aged < 2 months	Amphotericin B deoxycholate	1 mg/kg once daily	L-AmB	5 mg/kg once daily
			Caspofungin	25 mg/m^2^ once daily
			Micafungin	10 mg/kg once daily
Children aged ≥ 2 months	Caspofungin	70 mg/m^2^ once daily on day 1, followed by 50 mg/m^2^ once daily, (Max dose 70 mg)	L-AmB	5 mg/kg once daily
	Micafungin	2mg/kg once daily, in children ≥40 kg option to increase to 4 mg/kg once daily (Max dose 100 mg)		
**Age group**	**Prophylaxis in** **outbreak setting**	**Dosing**		
Neonates in NICUs <1000 g or who have risk factors for invasive candidiasis	Micafungin	3–4 mg/kg twice weekly or 2 mg/kg/day		

Refs. [111,126,127], L-AmB, Liposomal amphotericin B; NICU, neonatal intensive care unit. Of note, anidulafungin is not approved for use in children. ^1^ Alternative therapy in children aged ≥2 months should be considered in the instance of failure of first-line antifungal treatment, persistent candidemia (>5 days) or recent prolonged exposure (>4 weeks) to echinocandin class. Alternative therapy with echinocandins in neonates and infants aged <2 months should be used with caution and should only be considered if the central nervous system infection has been ruled out.

## Data Availability

Not applicable.

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
