# Peer review of "Navigating the New Reality: A Review of the Epidemiological, Clinical, and Microbiological Characteristics of Candida auris, with a Focus on Children"

_jof, 2023, doi:10.3390/jof9020176_

Round 1
Reviewer 1 Report
Abstract: It reflects accurately the essential information from the paper
The overall design of the study --- the narrative review was adequate and the methods adopted were adequately described.
The usefulness of the tables and figures--- The quality of the tables is low. The tables need improvement so that they would be interesting to readers.
Conclusion: The message was clear and sufficiently derived from the focus of the objectives. The article narrates the important associations of C. auris with nosocomial outbreaks in pediatric ICUs, invasive fungal infections, high-level antifungal resistance and high mortality rates. The authors highlight the importance of global collaborative efforts to raise awareness and limit its spread.
References: They were up-to-date and relevant to the article.
Recommendation
The article should be accepted after the correction of the layout for the tables
Reviewer 2 Report
Comments
In the introduction, reference to other relevant reviews on C. auris should be included such as
- Du H, Bing J, Hu T, Ennis CL, Nobile CJ, Huang G. Candida auris: Epidemiology, biology, antifungal resistance, and virulence. PLoS Pathog. 2020 Oct 22;16(10):e1008921. doi: 10.1371/journal.ppat.1008921. PMID: 33091071; PMCID: PMC7581363.
- de Cássia Orlandi Sardi J, Silva DR, Soares Mendes-Giannini MJ, Rosalen PL. Candida auris: Epidemiology, risk factors, virulence, resistance, and therapeutic options. Microb Pathog. 2018 Dec;125:116-121. doi: 10.1016/j.micpath.2018.09.014. Epub 2018 Sep 8. PMID: 30205192.
- Rhodes J, Fisher MC. Global epidemiology of emerging Candida auris. Curr Opin Microbiol. 2019 Dec;52:84-89. doi: 10.1016/j.mib.2019.05.008. Epub 2019 Jul 3. PMID: 31279224.
- Thatchanamoorthy N, Rukumani Devi V, Chandramathi S, Tay ST. Candida auris: A Mini Review on Epidemiology in Healthcare Facilities in Asia. J Fungi (Basel). 2022 Oct 26;8(11):1126. doi: 10.3390/jof8111126. PMID: 36354893; PMCID: PMC9696804.
Concerning section “5. C.auris antifungal resistance and therapeutic options”, i.e., the paragraph at line 372 on the use of antifungal susceptibility testing (AST): A suggestion is to add here a short discussion/review of the recently developed rapid AST methods and how these could help in selecting quickly a proper antifungal drug.
Minor comments
Abstract, line 25: Candida auris => C. auris
p. 3, line 96: C. Haemulonii => C. haemulonii
p.3, line 116: C auris => C. auris
p. 3, line 125: … (114, 45%) … (67, 26%) … => if 114 and 67 are referring to references, they should be surrounded by square brackets.
p. 3, line 129: the reference [66] is repeated.
p. 3, Figure 1 caption: Candida auris => C. auris
p. 13, line 300: United Stated => United States
p. 16, line 444: Figure caption should be under the figure.
